# Partner perceptions during brief online interactions shape partner selection and cooperation

Tiffany Matej Hrkalovic[1,2]*, Bernd Dudzik[2], Hayley Hung[2], Daniel Balliet[1]

1 Department of Applied and Experimental Psychology, Vrije Universiteit Amsterdam, Amsterdam, The Netherlands, 2 Faculty of Electrical Engineering, Mathematics and Computer Science, Delft University of Technology, Delft, The Netherlands

* t.matejhrkalovic@tudelft.nl

## Abstract

Evolutionary theory suggests that partner selection – the ability to identify and preferentially interact with individuals willing (warmth) and able (competence) to work towards mutual benefits – is a key driver of cooperative behavior. However, partner selection is complex, requiring the integration of various information, such as impression formation and task affordances. Despite its importance, there is limited research on the effect of these factors on partner selection for cooperative tasks. Thus, this paper investigates how person perceptions (warmth and competence), task affordances, and facial and acoustic nonverbal behavior inform partner selection for cooperative tasks. For this purpose, we asked participants to select partners for a task that either expressed warmth- or competence-related traits. Participants had a 3-minute (online) conversation with up to five individuals, reported their evaluations, selected partners for the task, and then engaged in the task. Results indicate that person perceptions guide partner selection, with each trait being more predictive in relevant tasks. Additionally, we found that the perceptions of warmth, but not competence, can be predicted by facial and acoustic cues during conversations. Lastly, we find that in the context of online social interactions, individuals were more cooperative towards selected participants than unselected. We discuss these implications in the context of the theory of partner selection and offer insights on how these results can be used in future efforts for designing socially intelligent artificial systems that support partner selection decisions.

## Introduction

Humans are inherently social beings. They collaborate on projects, share emotional support, and regularly engage in cooperative and interdependent tasks. Selecting suitable partners for these tasks is critical, as this decision can significantly affect task performance, satisfaction [1,2], well-being [3], and the level of cooperation achieved [4,5]. Yet, partner selection is a complex process, as individuals first need to evaluate if potential partners are willing and able to work collaboratively and if they meet the task demands [6,7]. Evaluating partners before selection is important as it can help people select the most suitable partner, thereby ensuring

**Data availability statement:** All data used in the paper is available in the OSF repository. The Questionnaire data and OpenSmile extracted features can be found via: https://osf.io/x8udn/ (DOI: 10.17605/OSF.IO/X8UDN). OpenFace extracted features can be found via: https://osf.io/j8efw/ (DOI: 10.17605/OSF.IO/J8EFW). All the files are linked to their respective documentation, as well as the codebook. The data can be found under Files in the OSF repository.

**Funding:** This study was supported by the European Research Council (https://erc.europa.eu/apply-grant/consolidator-grant) in the form of the Consolidator grant awarded to D.B. (864519) and Hybrid Intelligence Center funded by the Dutch Ministry of Education Zwaartekracht grant (024.004.022) (https://www.nwo.nl/onderzoeksprogrammas/zwaartekracht) in the form of salary for T.M.H. & B.D. and financing the data collection of the study.

**Competing interests:** The authors have declared that no competing interests exist.

mutually beneficial outcomes [6,7]. Consequently, these factors may be important in selecting partners for cooperative tasks. Yet, while most literature focused on exploring the role of these factors in romantic partner selection [8], less is known about the underlying mechanisms of partner selection for cooperative tasks. Therefore, this paper aims to investigate how individuals form partner evaluations and explore their role in predicting partner selection across various cooperative tasks.

When selecting partners, two key evaluation dimensions are important: the likelihood and ability of potential partners to act cooperatively or selfishly [4,6,9]. Prior research empirically highlighted the importance of proxies of these dimensions in partner selection decisions [10], especially in cooperative tasks that involve conflicts between individual and collective interests, known as social dilemmas [11,12]. Social evaluation models also emphasize these dimensions as the foundation for how people assess others. They are commonly referred to as warmth (willingness to cooperate) and competence (ability to contribute effectively) [13]. In everyday interactions, these perceptions are shaped by various social cues, such as non-verbal behaviors like smiling or voice pitch, which consistently enhance perceptions of warmth and competence in interactions with strangers [14–17].

However, several limitations persist in the current research investigating the underlying mechanisms of partner selection. Firstly, many psychology studies examining the link between non-verbal behavior and person perceptions, usually used non-interactive methods, such as photographs [10,16] or listening to isolated audio recordings [18], which fail to capture the dynamic nature of real-life interactions [19]. Secondly, most literature examined partner selection for social dilemma tasks [11,20], where selecting a warm partner is beneficial [12], but it overlooks tasks where partner competence has a greater impact on the outcome. To date, only a few studies have explored how task affordances influence partner selection, leaving it unclear how preferences shift based on whether partner warmth or competence are afforded to have a relatively greater impact on outcomes [21].

The current research addresses limitations of prior research by (1) investigating partner selection for cooperative tasks in real-life (online) social interactions, and (2) having people evaluate, select, and cooperate with partners on tasks that afford either warmth or competence to impact outcomes. This study makes three main contributions to the literature on partner selection and cooperation. First, this study explores the relationship between facial and acoustic nonverbal cues and perceptions of warmth and competence in brief online social interactions. Second, the study investigates how task affordances moderate the relation between person perceptions and partner selection. Third, the study evaluates whether partner selection is associated with successful cooperative outcomes.

## Person perceptions and non-verbal behavior

A key mechanism of partner selection lies in the ability to evaluate others, form perceptions, and use these perceptions to guide partner choice. These perceptions are shaped by a myriad of social cues [21,22], ranging from reputational information [4] to the partner's sex and age [24], social status [10,25], and many more. However, this paper focuses on non-verbal behavior cues [14,26]. These behavioral cues enable individuals to make quick, though sometimes imperfect, evaluations of others [16,27]. Nevertheless, these evaluations help individuals navigate social interactions by reducing uncertainty about unknown individuals [22].

Prior literature in social psychology has linked facial and acoustic cues, such as smiling, to perceptions of warmth [16,28–30]. While insightful, most of these studies, however, rely on non-interactive designs using static stimuli like photographs [16,28], videos [29], or isolated voice recordings to study partner perceptions [17,18,30] while only a small number of studies

explore these phenomena in real-life social interactions [31,32]. Using non-interactive stimuli limits the ability to capture the dynamic nature of real-life social interactions, such as capturing rapid changes in the speaker's pitch. Moreover, in social psychology, annotating this rapid change of non-verbal behaviors presents challenges, with manual annotation being the dominant but limited method. Manual annotation, where annotators review behavioral data and label specific non-verbal behavioral cues, is both time-consuming and inadequate for capturing subtle nonverbal behaviors (e.g., acoustic behaviors) and hard to use for tracking their rapid changes in behavior. On the other hand, automatic annotation, using algorithms powered by machine learning (ML), to analyze and label behaviors, offers a more efficient way to capture a wide range of nonverbal cues [33]. For instance, models like OpenFace [34] can track changes in facial movements through Action Units, while OpenSmile [35] analyzes acoustic features such as voice pitch, shimmer, and jitter. These tools provide a comprehensive and validated method for annotating nonverbal behavior, surpassing the limitations of manual annotation.

Research using models like OpenFace and OpenSmile has demonstrated that nonverbal behaviors, such as smiling, facial expressivity, and higher pitch, can predict perceptions of warmth [36,37]. However, studies linking non-verbal behavioral cues and perception of competence are scarce. A limited number of findings suggest that smiling, gesturing, and lower voice pitch positively influence perceptions of competence [15,37], while lower voice pitch and reduced facial expressivity are linked to perceptions of dominance [38,39], which is closely related to competence [40].

Despite advancements in automatic annotation, most studies focused on third-party perceptions (how observers perceive others) rather than first-party perceptions (how individuals in an interaction perceive each other). This distinction is important as it is well known that individuals who interact and observe the interaction have access to different information when forming perceptions [41].

The present research addresses these limitations by examining the relationship between automatically annotated facial and acoustic nonverbal cues and first-party perceptions of warmth and competence in social interactions. This study adopts an exploratory approach using machine learning to understand whether these cues are associated with person perceptions of warmth and competence.

## Person perceptions, partner selection, and cooperative behavior

Person perceptions are typically formed automatically and early in social interactions. While they can sometimes be inaccurate [16], these perceptions play a crucial role in guiding future decisions and behaviors [23,28], especially in cooperative contexts [11].

The two most important perceptions people use to evaluate others are warmth and competence [42]. *Warmth* represents a partner's willingness to engage in behaviors that benefit others, encompassing traits such as trustworthiness, kindness, and friendliness, while *competence* indicates an ability to act upon one's intention to benefit others, including skillfulness, knowledgeability, and intelligence [13,42]. These dimensions are also central to evolutionary models of partner selection, as they can inform a partner's willingness and ability to act cooperatively or selfishly [4,6,9]. This study integrates these frameworks by measuring social evaluation models of warmth and competence to examine their role in partner selection across different types of cooperative tasks in online social interactions.

In cooperative tasks, limited research has linked partner selection to perceptions of warmth and competence. For example, Raihani and Barclay (2016) found that individuals tend to choose fair (warm) and rich (competent) partners over unfair and poor partners in social

dilemmas. Similarly, Eisenbruch and Roney (2017) showed that people prefer to retain part-ners known for being generous (warm) and productive (competent). Both studies, however, relied on written reputational cues, such as prior behavior or shared cooperative history, which were devoid of actual social interactions between individuals. It remains unclear whether the same relationship would hold when warmth and competence must be inferred from subtler, less overt cues exchanged in typical social interactions with strangers. A study by Clark, Green, and Simons (2019) showed that people can assess warmth and competence from less overt cues in written statements and tend to select partners who score high on both dimensions. However, it is uncertain if similar perceptions guide partner selection within actual social interactions.

This study builds on previous research by investigating whether perceptions of warmth and competence still predict partner selection when based on more ambiguous cues, hypothesizing that both traits will positively influence partner choice. Additionally, we test whether partner selection enhances cooperative behavior, hypothesizing that individuals will be more coopera-tive toward selected partners than toward unselected ones.

## Situational affordances: trade-off between warmth and competence in partner selection

When selecting partners for cooperative tasks, individuals often prioritize warmth over com-petence, as a warm partner is seen as more likely to engage in behaviors that promote mutual benefit and collaboration [11,22] and can have a positive impact on individual reputation [43]. While competence—reflecting skills and intelligence—remains valuable, it is often viewed as secondary to warmth. However, cooperative tasks come with opportunities and barriers (i.e., affordances) for the expression of certain traits [44], where precedence of warmth over and beyond competence, can lead to sub-optimal partner selection when the cooperative task requires a competent partner for cooperative success. For instance, in tasks that require competence (e.g., a surgeon conducting a complex surgery), a partner who is trustworthy and friendly, but not competent, is a less suitable partner for this specific task. Thus, task affor-dances are critical for selecting suitable partners.

Research on the role of task affordances, though limited, indicates that people are sensi-tive to task affordances when selecting partners for cooperative tasks [21], and similar results have been found in different types of relationships [45]. For instance, Clark and colleagues (2019) found that individuals selected a highly competent partner for a task where they competed against another team but selected a highly warm partner in a task when they had to learn about partners in the other team. Though insightful, in this paper participants selected partners for either a competitive or collaborative task, rather than tasks requiring a tradeoff between warmth versus competence. Exploring this tradeoff in partner selection, Eisenbruch and Roney (2019) found precedence of warmth over competence in partner selection, but their method involved a task that afforded the expression of both warmth and competence, making it difficult to evaluate how the task affordance influenced partner selection.

It remains unclear how perceptions of warmth and competence influence partner selec-tion when the importance of these traits varies depending on the task. In this study, we explicitly manipulate task affordances by designing one task that prioritizes warmth and another that prioritizes competence. The competence-focused task involves aligned interests, requiring intelligence and skills for success. The warmth-focused task involves conflicting interests, where a partner's warmth is essential to ensure mutual benefit despite the tempta-tion to exploit. We hypothesize that individuals will prioritize high-warmth partners for the warmth-focused task and high-competence partners for the competence-focused task.

## Overview of the study

This paper investigates partner selection in cooperative tasks, focusing on: (1) how non-verbal behavior influences perceptions of warmth and competence, (2) the moderating role of task affordances on the link between partner perceptions and selection, and (3) whether partner selection impacts cooperative behavior and outcomes in online interactions.

To address these questions, we conducted a two-part online study. The two-part online study involved an initial survey where participants reported personality traits, availability, and broadband speeds. A week later, participants engaged in an interactive session in groups of four to six participants. During this session, participants took photos, evaluated others' photos, had a 3-minute online dyadic conversation with each other, rated each other post-conversation, and completed a cooperative task emphasizing either warmth or competence.

We took video and audio recordings of their initial conversations and then used OpenFace and OpenSmile to obtain automatic annotations of facial and acoustic behaviors. These annotations were fed into three machine-learning models to test their relationship with person perceptions of warmth and competence. Then, we used reported perceptions of warmth and competence to examine their role in partner selection decisions and tested whether these associations were moderated by task affordance. Lastly, we associated partner selection with cooperative behavior (i.e., the amount of money given to the other person) in one of the cooperative tasks.

We hypothesize that perceptions of warmth and competence will be positively associated with partner selection, with task affordances moderating this relationship - warmth being prioritized in warmth-related tasks and competence in competence-related tasks. We also expect that participants will be more cooperative with selected versus unselected partners. All the hypotheses were preregistered (https://osf.io/u9k58/?view_only=1acfce82cb504b22b722a-3bafe37d4cf), while the analysis of the association between the automatically annotated facial and acoustic cues and perceptions of warmth and competence was exploratory.

## Method

### Participants

Participants were recruited from the UK via the online platform Prolific (http://www.prolific.co). A total of 297 participants completed both parts of the study. Due to the study's complexity, no a priori power analysis was conducted; the sample size of 300 was determined based on the available time and budget. The final number of participants decreased to 279 participants (Female = 154; Male = 122; Non-binary = 2; Unknown = 1; $M_{age}$ = 36.69, $SD_{age}$ = 11.03; $Range_{age}$ = 19–67 years old) after excluding participants who dropped out or had incomplete self-reported data.

Each participant interacted with four to six other participants, resulting in 1383 photograph ratings, 1385 conversations, and 1326 cooperation reports. Observations were further reduced to 1136 by excluding incomplete interactions and those with less than two minutes of conversation. The threshold of two minutes was used, as individuals could stop the conversation at two minutes by hanging up the call. Thus, all the conversations that lasted less than two minutes finished earlier due to some technical complications and were not taken into consideration for further analysis. After removing observations with missing survey data or recordings, 1080 interactions were included in the analysis. For the machine learning component, 1022 observations were used, as some videos were unsuitable for facial feature extraction. Additional details about the sample, including nationality and inclusion criteria, are available in the S1 File.

## Procedure

The study consisted of two parts: an intake session and an interaction session. Initially, participants were informed that the research focused on social decision-making in everyday interactions, involving real-time interactions with others. Participants were presented with a consent form detailing the study's scope, earnings, bonus payments, and their rights, including the option to share anonymized data with other researchers. After providing written consent by answering questions to participate and allowing us to share their anonymized data they continued with the rest of the intake session. In the intake session, participants completed a 40-minute online survey measuring variables relevant to the experiment, such as social anxiety, personality traits, and intelligence (see Measures). They also scheduled a time for the second part of the study. At the end of the study, they were debriefed and told that a researcher would send a message on Prolific with details about the second part of the study.

Between the intake and interactive session, participants received a Prolific message detailing what to expect in the second part of the study. They were instructed to participate in a quiet environment with a laptop, headphones, and a microphone. Participants without the required equipment were excluded from the study. Researchers also provided a link for accessing the interactive session.

About a week later, using the provided link, participants joined a four-to-six-person round-robin interactive session. Ten minutes before their scheduled slot, participants started to enter the study. During these ten minutes, researchers conducted brief video calls with every participant to ensure participants had the necessary equipment and environment. Participants with satisfactory conditions were then directed to a waiting room, and those who did not meet the requirements were redirected to a different study, which is outside of the current scope of this paper.

The interaction session began with participants receiving instructions on the experiment flow and the cooperative task. Each group of participants was randomly assigned to either select a partner for a competence-focused or warmth-focused task. Note that participants were not told that they needed to select partners that are high on warmth or competence. Importantly, the word cooperative was not used, rather both tasks were presented as "Decision tasks" (see Experimental Tasks). The interactive experiment involved four stages. In the first stage, participants took a photo with their web cameras and then saw photos of others and rated them on warmth and competence. When they saw all the photos, again photos were presented in a successive order followed by a question querying if they wanted to do the decision task with this participant or not. After the first partner selection, participants engaged in 3-minute dyadic conversations with each group member. Each conversation was followed by a survey to rate their conversation partner. After all the conversations, photos of all participants were presented in a successive order followed by a question querying if they wanted to do the cooperative task with this participant or not. After the post-conversation partner selection, participants completed a warmth-oriented or competence-oriented task with each group member, regardless of their partner selection. Here, participants did not interact in real time, rather photographs were used to identify their partners. Participants were not aware that they would be paired with all participants, but the researcher said that their selection would be important for calculating their bonus payments (for more detail see S1 File). After the cooperative task, participants engaged in a collaborative task, which is not part of this study, so we will not describe it here (for a full visualization of the interaction session see Fig 1).

At the end of the session, participants uploaded their video-audio recordings and were debriefed. The entire study, conducted in Qualtrics, took place between April 19th and July 14th, 2022. This research was approved by the Ethical Board of the VU Amsterdam

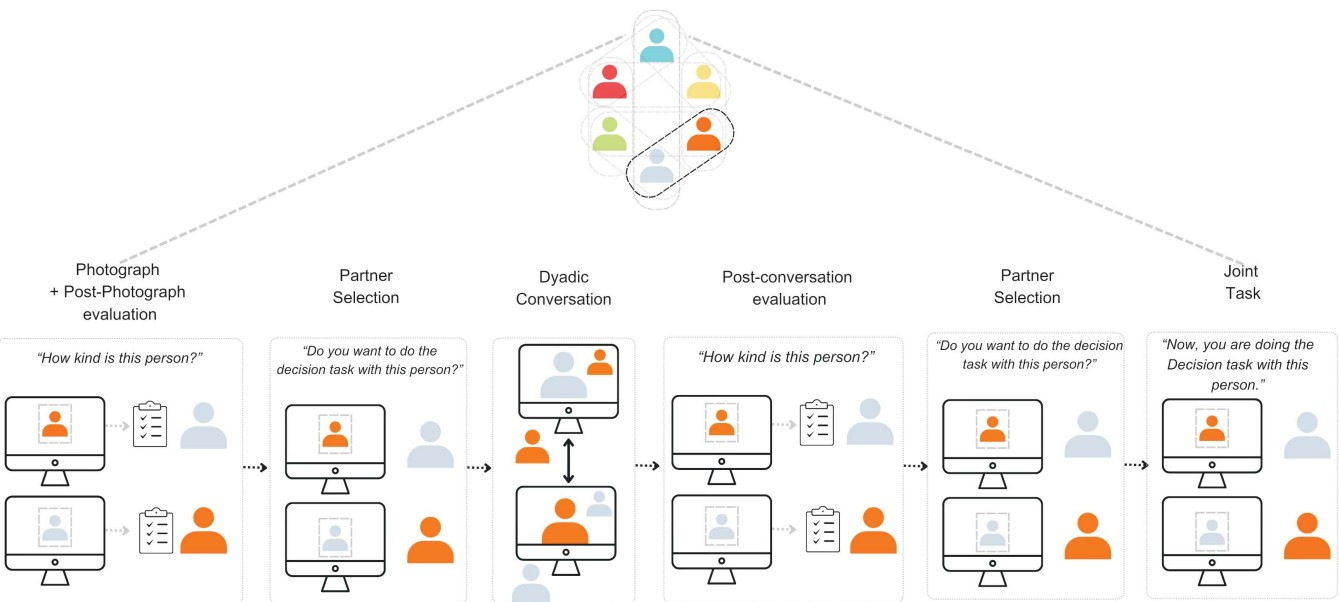

**Fig 1. Visualization of the experimental setting of the Interaction session.** The Interaction Session followed one week after the Intake Session. Here, each participant was placed within a group (4-6 participants). Each participant interacted with all other participants in their group. Every participant repeated every stage (Photograph + Evaluation of Person Perceptions, Partner selection, Dyadic conversation + Post-conversation evaluation, Partner Selection, and the Joint Task) p-times (p = number of participants in their group - 1)). For each evaluation, partner selection, and decision in the Joint Task, participants were presented with a picture of the participant they would be rating, selecting, or interacting with (for more information see Procedure). This figure represents the experimental procedure experienced by two participants (black lines), while all participants interacted with each other (grey lines).

(VCWE-2021–168). All participants provided written consent to participate and share anonymized data.

## Measures

In the intake session, participants completed an online survey measuring various traits relevant to task outcomes, such as broad personality traits, social anxiety, psychopathy, and social value orientation. In the interaction session, participants evaluated each other on warmth, competence, and other characteristics.

**Intake session.** *Personality, psychopathy,* and *social anxiety.* Personality traits were measured using the HEXACO-60 [46], which assesses six dimensions: Honesty-Humility ($\alpha$ =.75), Emotionality ($\alpha$ =.78), Extraversion ($\alpha$ =.83), Agreeableness ($\alpha$ =.83), Conscientiousness ($\alpha$ =.80), and Openness to Experience ($\alpha$ =.80) through 60 items (e.g., "*I tend to be lenient in judging other people.*"). Psychopathic traits were assessed with the 20-item Psychopathic Personality Traits Scale (PPTS) [47] (e.g., "*What other people feel doesn't concern me.*"; $\alpha$ =.86). Social Anxiety was measured using the 6-item Social Interaction Anxiety Scale (SIAS-6) [48] (e.g., "*I have difficulty making eye contact with others.*"; $\alpha$ =.84). All items used a 5-point Likert scale (1 = "*Strongly disagree*" to 5 = "*Strongly agree*").

*Social value orientation.* Social Value Orientation (SVO) was measured using the Slider Measure [49], which evaluates how individuals value their own and others' outcomes. Participants completed six tasks, each requiring them to distribute a fixed amount of money between themselves and an anonymous individual. The overall SVO score was computed based on their distributional decision.

*Trustworthiness and ability.* Trustworthiness and ability were assessed using the Benevolence ($\alpha$ =.84) and Ability ($\alpha$ =.80) sub-scales from Mayer et al.'s (1995) [50] Trust model. Each

dimension was measured with a 5-item scale on a 5-point Likert scale (e.g., Benevolence: "I am very concerned about others' welfare"; Ability: "I am known to be successful at what I try to do").

***Non-verbal intelligence and reasoning.*** Non-verbal intelligence was measured using the University of California Matrix Reasoning task (UCMRT) [51]. The UCMRT consists of 23 matrix test problems, two example problems, and six practice problems that participants needed to solve in 10 minutes.

**Interaction session.** *Person perceptions*. Person Perceptions were assessed using established scales for warmth (comprising sociability and morality), competence [52], similarity, and physical attractiveness. Warmth was measured with a six-item scale: three items assessed sociability (e.g., "How sociable/friendly/kind is this person?"), and three items measured morality (e.g., "How trustworthy/honest/sincere is this person?"). All subscales demonstrated high internal consistency (Warmth: $\alpha_{photo}$ =.91, $\alpha_{conv}$ =.90; Sociability: $\alpha_{photo}$ =.81, $\alpha_{conv}$ =.83; Morality: $\alpha_{photo}$ =.91, $\alpha_{conv}$ =.90). Competence was assessed with items like "How intelligent/skillful/competent is this person?" ($\alpha_{photo}$ =.91; $\alpha_{conv}$ =.91). Physical attractiveness and similarity were measured with single items (e.g., "How physically attractive/similar in beliefs is this person?"). Perceptions were recorded on a 7-point Likert scale after viewing a photo and after each conversation.

***Partner selection***. Partner selection was measured with one item after viewing photographs, and then again after each conversation (e.g., *"Would you want to do the decision task with this person?"*). Participants provided a binary response (no, yes).

***Cooperative Behavior.*** Cooperation was measured as the amount of MU participants gave to each other in the Joint Trust Task (see Experimental Tasks).

## Experimental Tasks

At the start of the Interaction Session, participants read task instructions and watched a video specific to their group's tasks. They were then familiarized with the task. Afterward, they answered three comprehension questions about the task.

***Joint Competence Task***. The Joint Competence Task is an interdependent activity where participants could mutually benefit. Each participant received an initial endowment based on their performance in the UCMRT intelligence test during the intake session (0.50 pounds per point, up to 11.5 pounds). The endowments were pooled, with each dyad having its own shared pool. To win the pooled money, participants had to solve an additional UCMRT problem. Outcomes were: 1) If both answered correctly, they split the pool evenly; 2) If only one answered correctly, each received 1/4 of the pool; 3) If both answered incorrectly, neither received anything. The task emphasized competence over warmth, as participants earned more when both correctly solved the problems. In summary, in the joint competence task, people are interdependent with corresponding interests, where individuals earn more money when both themselves and their partner correctly solve a problem. Hence, this task requires the partner's competence to influence outcomes, while partner warmth should not impact task outcomes.

***Joint Trust Task***. The Joint Trust Task is a modified prisoner's dilemma where participants each receive a 10 MU endowment (0.50 pounds per MU). Participants decide how much to allocate to their partner (0–10 MU), with any amount given being worth 1.2 times more to the partner. Maximum profit is achieved by keeping all MUs while receiving the partner's full endowment, creating an incentive to exploit. However, both participants achieve a better outcome by giving their entire endowment to each other (6 pounds) compared to keeping it (5 pounds). This task highlights interdependence and conflict of interest. For more details, see S1 File, Note 2.

**Audio-video recordings.** Audio-video recordings were recorded using web cameras. All videos were standardized to 640x480 resolution at 30 frames per second (fps). Each recording captured only one person (see Fig 2 for the modeling process). Before extracting behavioral cues, recordings were converted from.webm to.mp4 format using *ffmpeg* [53]. Audio tracks were extracted from video-audio recordings, saved as.wav files, and re-sampled to match the video's frame rate (30 fps). Only videos longer than two minutes were analyzed, as participants could end the call after that time.

**Facial and acoustic nonverbal behaviors.** Frame-level facial and acoustic nonverbal behavior were automatically extracted using open-source algorithms. To do this, audio and video recordings of the initial conversations were used as input to these algorithms.

Facial cues were obtained from video recordings using OpenFace [34], which provided intensity and occurrence data for action units (AU), head orientation, eye gaze, and facial landmarks (see S1 File, Note 3 for the full list). Here, only intensity of AUs, head orientation, and eye gaze were used.

Acoustic cues were extracted from audio recordings with OpenSmile [35], specifically using the CoMpAre2016 feature set, which includes 64 low-level descriptors per frame (e.g., pitch, jitter, shimmer). To synchronize facial and acoustic data, features were recorded every 33.3 milliseconds. For a full list of features see S1 File, Note 3.

After extractions, conversations shorter than three minutes were zero-padded (adding zero values) to ensure consistent time-series lengths of up to 5400 time points (approximately 3

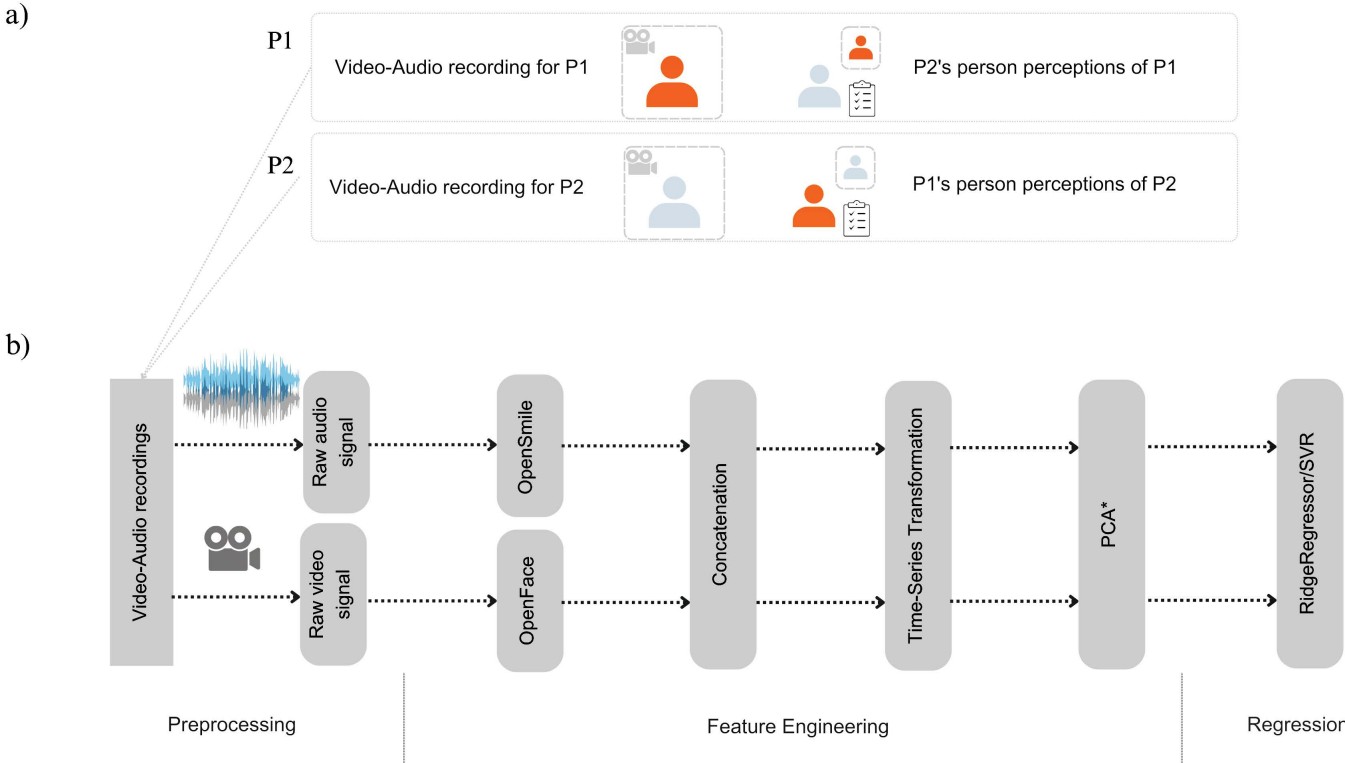

**Fig 2. Machine Learning Modeling Setup.** The figure represents: a) the data pairing used for machine learning analyses with time-series features of facial and acoustic cues automatically annotated in videos of each participant as input, while output were ratings of these participants on warmth or competence by others; b) presents the analysis process from multi-modal data preprocessing, where each audio-video recording was preprocessed by separating video and audio streams, fed to a pre-trained model used for vocal (OpenSmile) and facial (OpenFace) cues extraction. Lastly, Principal Component Analysis was only applied to reduce the dimensionality of the input that was used as input for the Support Vector Regression (SVR).

minutes). Each annotated nonverbal behavior from an audio and video recording was linked to a specific time series. These synchronized time series were then analyzed to capture relevant patterns and signals.

## Preprocessing: representing multivariate time-series of non-verbal behavior with minirocket

Before reporting the results, we will detail the preprocessing and modeling procedures to provide clarity on our methods that took place in preparing the machine learning pipeline to analyze the relationship between non-verbal facial and acoustic cues and person perceptions of warmth and competence. This will include a description of how the nonverbal behaviors were prepared for analysis, as well as an overview of the modeling techniques used in these analyses.

Firstly, instead of averaging the time series. we used an approach that transforms time series by capturing the characteristics of the time series of facial and acoustic cues. This is done by the MiniROCKET algorithm (Minimally Random Convolutional Kernel Transformation), which preserves the dynamic nature of behavioral cues by capturing morphological properties of the raw time series, such as the prevalence or absence of peaks or valleys within a specific shape. MiniROCKET (MINImally RandOm Convolutional KErnel Transformation) is a relatively recent approach for time-series transformation or feature extraction [54] (for more details on MiniROCKET and configuration used, see S1 File, Note 12). Here, we applied the MiniROCKET Multivariate variant to transform each high-dimensional matrix of times series data into a single vector. This variant handles multiple time series of behavioral cues and not only captures the prevalence of morphological patterns within a specific time series for an individual behavior signal but also across the different variables involved.

The input to the MiniROCKET algorithm was a highly dimensional matrix where each video-audio recording of each conversation ($n_{videos}$ = 1022) was represented as 5400 rows (timepoints) x 114 columns (OpenSmile ($n_{columns}$ = 65) and OpenFace features ($n_{columns}$ = 49) (see Fig 2). After the transformation, each audio-video recording was associated with a row vector of 9996 MiniROCKET Multivariate time series characteristics. These time-series features were further used for a machine-learning algorithm.

**Modeling process.** In line with the MiniROCKET and prior recommendations [54] time series transformations were used as input for a Ridge Regression model, which includes a parameter for reducing the risk of overfitting and facilitates fitting data with more predictors than observation-*regularization*. Ridge Regression assumes a strictly linear relationship between the predictors (time-series features of facial and acoustic cues) and the output variable (partner's perceptions of warmth/competence). To explore the potential benefits of a non-linear relationship, we also used Support Vector Regression with a *Radial-Basis Function* (RBF) kernel [55]. SVMs (Support Vector Classifier and Regressor) have been used in similar experimental settings [64]. To test whether our models performed significantly better than chance, we compare them with a baseline model (Dummy Regressor), that always predicted the mean of the target variable measured on the training dataset.

To evaluate model performance and select the best hyperparameters, we employed a 5-fold nested cross-validation procedure (for more details see S1 File, Note 13). Hyperparameters are settings of the model that are adjusted by the researcher and are not learned during the training process. The training process was done on 80% of the data (four of the folds), which was used as the training set, and 20% (the remaining fold) was used as the test set. Given the nested nature of the experimental setup—where participants only interacted with others in their group— we ensured that data from the same group remained within the same fold to

avoid information leakage. Consistent train-test splits were used across all models to facilitate fair comparison. The training set was further divided into development and test sets for hyperparameter tuning, with the best combinations identified through grid search and the 5-fold inner splitting of the development set. Model performance was evaluated using the $R^2$ score, and all analyses were conducted using the Python library Scikit-Learn [56].

## Results

To control for the false positive rate due to multiple comparisons, all *p*-values were adjusted using the Benjamini & Hochberg (1995) [57] adjustment method controlling for false discovery rate. The adjusted p-values were calculated using the *stats* package in R (version 4.3.2).

### Nonverbal cues and person perceptions of warmth and competence

To explore the relationship between person perceptions of warmth and competence and facial and acoustic non-verbal cues during brief online conversations, we conducted a machine learning analysis. We used three models—Dummy Regressor, Ridge Regressor, and Support Vector Regressor (SVR)—for each output variable (warmth and competence).

Comparison between the models showed that the Ridge Regression model achieved the best performance in predicting perceptions of warmth. The Dummy Regressor underperformed showing a negative performance metric ($R^2$ = -.02). Results show that the combination of acoustic and facial cues managed to predict and explain 5.9% of the variance of perceptions of warmth (see Table 1). Running the t-test comparisons between model performances, it was shown that the Ridge Regression achieved significantly better performance than the Dummy Regressor ($t(4)$ = 10.56, $p$ =.001, $d$ = 4.23). However, this was not the case for models using only facial ($t(4)$ = 1.69, $p$ =.166, $d$ = 0.75) or only acoustic ($t(4)$ = 2.37, $p$ =.077, $d$ = 1.81) nonverbal cues. SVR was also significantly better than the Dummy regressor ($t(4)$ = 3.09, $p$ =.037, $d$ = 1.96), explaining 4.4% of the variance of perceptions of warmth. However, as the Ridge Regressor had the best performance, we are only reporting the statistics of the best-performing model in Table 1.

Facial and acoustic cues were not significant predictors of person perceptions of competence, as the Ridge Regressor model ($R^2$ = -.003) and SVR ($R^2$ = -.01) did not significantly outperform the Dummy Regressor ($R^2$ = -.02).

Note that here, the performance metric reflects how well the fitted model predicts test data—data not used during model training. It indicates the model's ability to generalize the relationship between nonverbal behavior and perception of warmth to new participants.

### Person perceptions, task affordances, and partner selection

To test the hypotheses related to the role of person perceptions and task affordances in predicting partner selection, two mixed-effects logistic regression models were fitted. The first

**Table 1. The proportion of explained variance ($R^2$) in the Ridge Regression by acoustic and facial modalities on unseen participants.**

| Variable | Modality of behavioral cues | | | | | |
|---|---|---|---|---|---|---|
| | Acoustic | | Facial | | All | |
| | $R^2$ | SD | $R^2$ | SD | $R^2$ | SD |
| Person perception of Warmth | .029 | .03 | .028 | .08 | .059*** | .02 |
| Person perception of Competence | – | – | – | – | -.003 | .01 |

Note: *** Models that were significantly better than the dummy model with the mean

model contained all control variables, while the second model included warmth and competence perceptions, along with their interactions with task type, as fixed effects predicting partner selection. We controlled for physical attractiveness, perceived similarity, interaction order, batch size, and task type (photography vs. conversation). Batch size indicated the number of participants that were in each group. The batch size could vary between four to six participants (see Method). The perceived similarity and attraction were controlled due to prior findings showing that these dimensions also affect preferential assortment [58,59]. Additionally, to account for subject and partner effects, subject and partner identification numbers were modeled as random effects. In each model, adding a dyadic effect [60] resulted in overfitting, indicating that the effect did not account for any additional variance above the subject effect and was thus removed. Therefore, only subject and partner random effects are used. For detailed descriptive statistics on person perceptions from the photographs and conversations, as well as changes in perception, see S1 File, Note 5 and Note 6.

In the control model, individuals perceived as more attractive ($b = 0.60$, $SE = 0.08$, $OR = 1.82$, $z = 7.38$, $p <.001$) or similar in beliefs ($b = 0.97$, $SE = 0.09$, $OR = 2.64$, $z = 11.34$, $p <.001$) were more likely to be chosen as partners. People were also more discerning in their partner selection after conversations compared to when viewing photographs, with higher selection likelihood from photographs ($b = 0.62$, $SE = 0.12$, $OR = 1.85$, $z = 4.93$, $p <.001$). The control model accounted for 27.14% of the marginal variance ($R^2cond = 50.16\%$) of partner selection. Neither the order of conversation and images ($b = -0.02$, $SE = 0.04$, $OR = 0.98$, $z = -0.44$, $p =.660$) nor batch size ($b = -0.21$, $SE = 0.16$, $OR = 0.81$, $z = -1.34$, $p =.181$) were significant predictors (for tables, see S1 File, Note 8).

When including the hypothesized predictors (warmth and competence perceptions and their interactions with task type) in the model, these factors explained an additional 9.70% of the marginal variance in partner selection. As anticipated, individuals perceived as warmer ($b = 0.57$, $SE = 0.15$, $OR = 1.77$, $z = 3.75$, $p <.001$) and more competent ($b = 1.00$, $SE = 0.15$, $OR = 2.72$, $z = 6.69$, $p <.001$) were generally more likely to be selected as partners. There was a significant interaction effect between task type and perceptions of warmth, and type of task and perceptions of competence. Specifically, warmth was a stronger predictor in the Joint Trust Task compared to the Joint Competence Task ($b_{JTT} = 0.38$, $SE = 0.19$, $OR = 1.46$, $z = 2.07$, $p <.001$). On the other hand, competence was less predictive of partner selection in the Joint Trust Task, compared to the Joint Competence Task ($b_{JTT} = -1.20$, $SE = 0.20$, $OR = 0.30$, $z = -6.04$, $p <.001$) (see Fig 3). Batch size ($b = -0.11$, $SE = 0.17$, $OR = 0.90$, $z = -0.62$, $p =.533$) and ordering ($b = -0.00$, $SE = 0.04$, $OR = 1$, $z = -0.01$, $p =.989$) did not significantly predict partner selection, nor did task type ($b_{JTT} = 0.13$, $SE = 0.22$, $OR = 1.14$, $z = 0.58$, $p =.561$).

Simple effects revealed that for the Joint Trust Task, only warmth had a significant positive association with partner selection ($b = 0.87$, $SE = 0.16$, $OR = 2.37$, $z = 5.49$, $p <.001$), while competence did not predict selection ($b = -0.27$, $SE = 0.14$, $OR = 0.76$, $z = -1.94$, $p =.052$). Conversely, in the Joint Competence Task both warmth ($b = 0.68$, $SE = 0.17$, $OR = 1.97$, $z = 4.00$, $p <.001$) and competence ($b = 1.03$, $SE = 0.15$, $OR = 2.80$, $z = 6.67$, $p <.001$) had positive significant associations with partner selection.

Given that the Joint Trust Task emphasized morality (trustworthiness) over general sociability (kindness and friendliness), we conducted an additional supplementary and exploratory analysis separating these facets. Results indicated that sociability ($b = 0.62$, $SE = 0.16$, $OR = 1.86$, $z = 3.91$, $p <.001$) was a significant predictor of partner selection, while morality was not ($b = -0.12$, $SE = 0.20$, $OR =0.89$, $z = -0.62$, $p =.536$). However, participants perceived as high in morality were more likely to be selected for the Joint Trust Task ($b_{JTT} = 0.66$, $SE = 0.25$, $OR = 1.93$, $z = 2.67$, $p =.008$), whereas sociability did not impact selection differently in two tasks ($b_{JTT} = -0.14$, $SE = 0.21$, $OR = 0.87$, $z = -0.68$, $p =.498$). For further details, see S1 File, Note 9.

While person perceptions of warmth and competence were predictive of whom individuals select as a partner, a supplementary analysis showed that there is no alignment between other's perceptions of warmth and self-reports of social value orientation, agreeableness, extraversion,

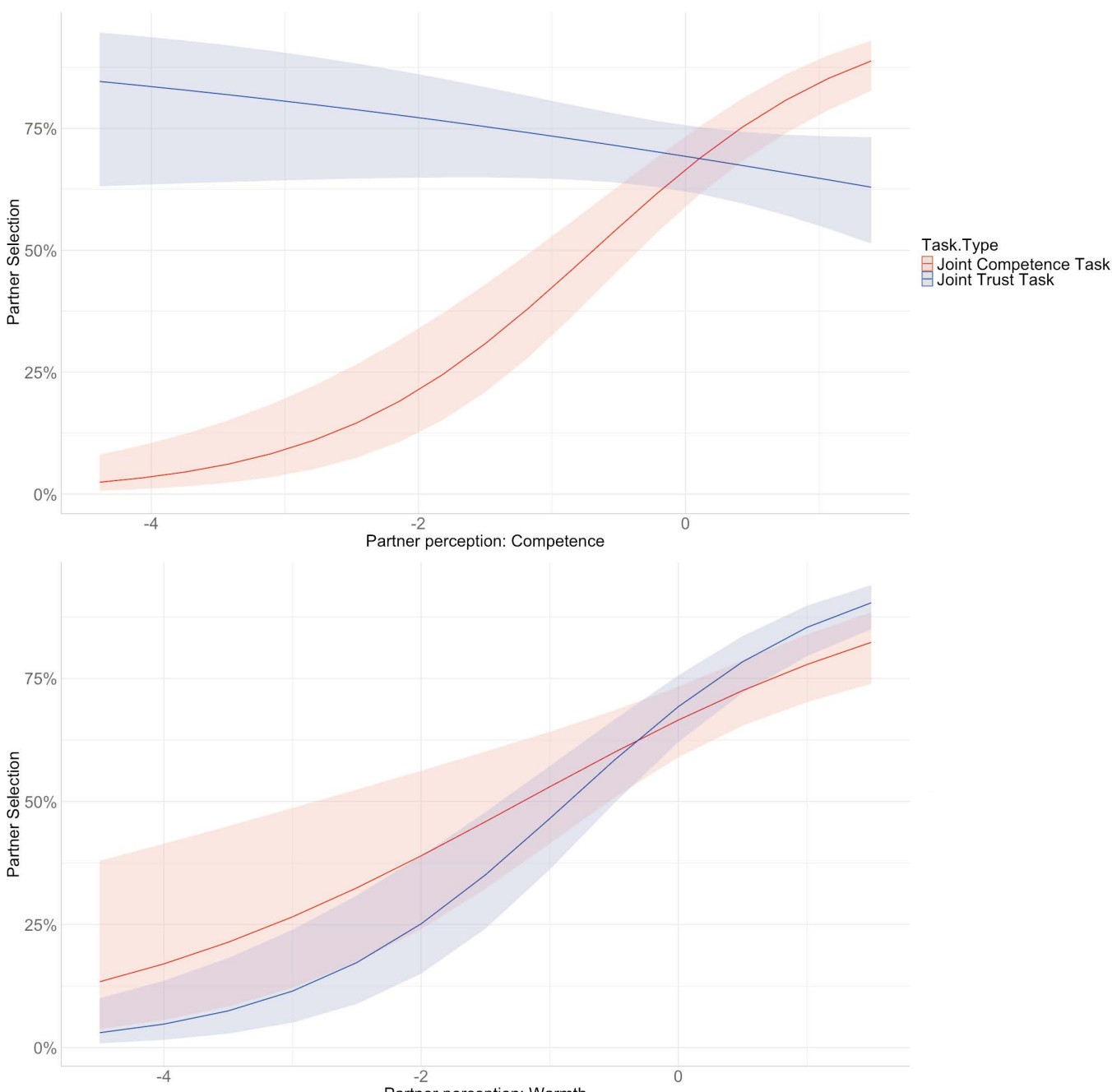

**Fig 3. Interaction Between the Type of Task and Person Perceptions of Warmth and Competence Predicting Partner Selection.** The figure illustrates how the type of task moderates the relationship between perceptions of warmth and competence in predicting partner selection. Specifically, participants perceived as higher in competence were more likely to be selected as partners for the Joint Competence Task, but this relationship was not observed for the **Joint** Trust Task. Participants perceived as higher in warmth were more likely to be selected as partners for both tasks. However, partner selection was slightly stronger for the Joint Trust Task compared to the Joint Competence Task.

honesty-humility, psychopathy, social anxiety, or benevolence. However, success in the UCMRT abstract task did positively predict how those individuals were perceived on competence, but self-evaluations of ability did not. For more details, see S1 File, Note 11.

## Partner selection and cooperation

Lastly, we investigated the role of partner selection in cooperative behavior as measured in the Joint Trust Task. The cooperative behavior was measured through a proxy of the amount of MU sent in the Joint Trust Task. Here, we only focused on partner selection after conversations ($n$ = 563 observations, $n$ = 138 participants). We used a robust linear mixed model with Huber-White sandwich standard error estimators due to deviations from normality and homoscedasticity. Cooperative behavior measured through the MU given was the outcome variable, and partner selection was the fixed effect, while including subject ID as a random intercept.

As predicted, participants allocated more MU to selected partners ($M$ = 6.98, $Mdn$ = 8, $SD$ = 3.36) than to non-selected partners ($M$ = 4.49, $Mdn$ = 5, $SD$ = 3.41) ($b_{selected}$ = 2.20, SE = 0.17, $t$ = 12.99, $p$ <.001). The model explained 9.10% of the marginal variance in cooperation, 7.40% more compared to the control model that included only control variables (see S1 File, Note 10).

Additionally, to investigate whether individuals were good at selecting partners who cooperate in the Joint Trust task, we also considered whether partners who were perceived to be higher on warmth and were selected, engaged in more cooperative behavior in the Joint Trust Task. Perceptions of a partner's warmth after the online interaction were not associated with the partner's cooperative behavior ($b$ = 0.15, $SE$ = 0.10, $t$ = 1.51, $p$ =.132). However, selected partners were only marginally more cooperative than unselected others ($b$ = 0.36, $SE$ = 0.18, $t$ = 2.01, $p$ =.045). These results indicate that while participants use their perceptions to inform their partner selection and cooperation, these person perceptions might not be very predictive of the partner's actual cooperative behavior, while partner selection is marginally associated with cooperative behavior. For more details see S1 File, Note 10.

## Discussion

This study examined how individuals form perceptions of warmth and competence during brief online social interactions, and how these evaluations were used to select partners for cooperative tasks that differed in their affordance for warmth- and competence-related traits. Furthermore, we assessed the consequences of partner selection on cooperative behavior. Video and audio recordings were used to analyze nonverbal cues during the social interactions, which weakly predicted perceptions of warmth but not competence. Participants preferred partners they perceived as both warm and competent, with task affordances moderating selection: in the competence-based task (i.e., the Joint Competence Task), participants favored competent partners, while in the warmth-based task (i.e., the Joint Trust Task), warm partners were more preferred. To closely inspect the preferences for a warm partner, we wanted to further separate the two subdimensions of warmth, namely, sociability and morality, and investigate their role in the Joint Trust Task. We found that people displayed a stronger preference for a highly moral (trustworthy) partner in the Joint Trust Task, compared to the Joint Competence Task.

Despite the influence of perceived warmth and competence on partner selection, supplementary analyses showed that these perceptions did not align with relevant self-reported traits. Moreover, results showed that participants who were perceived to be warmer did not act more cooperatively in the Joint Trust Task. On the other hand, being selected had a very

small effect on predicting a partner's cooperative behavior. Nevertheless, participants were more cooperative with the partners they selected, compared to unselected partners, which supports the notion that partner selection can facilitate cooperation.

## Theoretical implications

Perception of individuals' warmth and competence is crucial in partner selection [6], yet there is a notable gap in understanding how these perceptions form. Previous research has established a connection between nonverbal behaviors—such as smiling and gesturing—and perceptions of trustworthiness and approachability [15–17,36,37]. However, there is limited literature on how automatically annotated facial and acoustic nonverbal behaviors predict perceptions of warmth and competence from a first-person perspective. Our study addressed this gap, finding that automatically detected acoustic and facial cues during conversations were predictive of perceptions of warmth, but not competence. The inability to train a model to predict perceptions of competence using only facial and acoustic features does not support the findings of prior work that investigated the role of pitch on perceptions of prestige and dominance [17,38,39]. However, the inability to predict perceptions of competence could be due to the conversational settings. Indeed, initial conversations usually afford for behaviors that are mostly indicative of one's warmth, while competence is usually detected in behaviors where one's intelligence can be measured (e.g., task performance) which were not afforded in the short-term conversations. Moreover, participants couldn't engage in *cheap talk* [61], as we did not communicate to them how well they did in the intelligence tasks from the Intake session. Thus, making it harder for them to verbally communicate their intelligence. Similarly, we focused on a limited set of facial and acoustic non-verbal behaviors. For instance, previous literature has highlighted the significance of other conversation characteristics, such as turn-taking or speaking length, in predicting perceptions of dominance [39], which is associated with competence. Therefore, subsequent research may consider including other conversational characteristics when investigating perceptions of competence.

In the case of perceptions of warmth, the Ridge Regressor model performed significantly better than the Dummy Regressor but managed to explain only 5.9% of the variance of perceptions of warmth when generalized on the holdout set (e.g., participants that were not included in the training phase). This indicates that the generalizability of the model is somewhat limited. However, this might be due to the setup and input used. For instance, we used a hard setting for data training, using the perfect cut-off between training and testing data to limit data leakages. However, given the high subjectivity of person perceptions, the generalizability of the relationships between nonverbal behavior and person perceptions of unknown people might be a difficult task for the model. Prior literature showed that changes in distributions of training and test data could help in increasing model performance when detecting speech activity from body movement [62]. Nevertheless, our findings align with recent studies [63], who reported similar explanatory power ($R^2 = .08$) for perceptions of extraversion using nonverbal cues in non-interactive hiring interviews. Although our focus was only on nonverbal cues, Koutsoumpis and colleagues (2024) [63] found that the inclusion of verbal information significantly improved the model's explanatory power ($R^2 = .40$). Thus, future research should investigate this, by adding text data, that captures the content of conversations, in the models predicting person perceptions of warmth and competence. Our machine-learning approach offers several advantages over previous research. First, we focus on person perceptions from a first-party perspective in an online setting, a perspective explored by a few studies [36]. We trained our models using time-series patterns of facial and acoustic cues, a method not previously applied to studying warmth and competence perceptions in cooperative

settings. However, this approach has proven useful in modeling similar interactions [64]. Second, we demonstrate that automatically extracted non-verbal behaviors can predict how individuals are perceived in social interactions, but future research should address additional social cues to improve predictive performance.

Understanding whether machine learning can predict warmth and competence is important for two reasons. First, it provides new methodologies to model and analyze complex multivariate temporal processes between non-verbal cues of facial and acoustic behavior and social perceptions of warmth and competence. Second, it provides insights for fields like AI (Artificial Intelligence) in exploring whether models trained on non-verbal behaviors can predict abstract social phenomena, such as evaluations of potential partners. This could bring us closer to developing socially aware systems that assist in partner selection and promote cooperation [65,66].

Perceptions of warmth and competence are believed to influence partner selection, with prior research indicating a preference for warmth over competence when selecting partners for hypothetical scenarios [11,20]. Our study extends this understanding by examining these perceptions as they develop during social interactions. We found that individuals tend to choose partners perceived as both warmer and more competent. However, task affordances can moderate this relationship. In the Joint Competence Task, participants preferred more competent partners, while in the Joint Trust Task, they favored partners perceived as warm (trustworthy). However, warmth was also a strong predictor of partner selection in the Joint Competence Task. Further analysis revealed that in the Joint Trust Task, competence was not a significant predictor of partner selection, whereas warmth was a strong predictor in both tasks, with a higher predictive weight in the Joint Trust Task. These results are in line with prior literature showing the difference in partner selection preferences for cooperative and competitive tasks [21], as well as social preferences in different relationship types [45,67]. For instance, one study [21] observed that preferences for competence increase in relationship types affording competence, such as study or business partnerships. However, warmth-related traits such as trustworthiness and cooperativeness were preferred in all relationship types, including friends or romantic partners. Findings showing people's high preferences for warm partners in both tasks are also in line with other existing literature showing the primacy of warmth over competence in social perceptions and partner selection [11,12]. Arguably, morality (i.e., trustworthiness) is the most relevant trait for the Joint Trust Task, and indeed we found that when only doing analyses with the morality items, perceptions of morality were more predictive of partner selection in the Joint Trust Task, relative to the Joint Competence Task.

Furthermore, while we found that perceptions of warmth and competence were predictive of partner selection, the supplementary analysis showed that these perceptions were not completely aligned with how participants personally saw themselves. While not in line with our hypotheses, this finding is in line with literature questioning the accuracy of person perceptions, where most evidence indicates that people tend to overestimate other's traits during social interactions [16,27]. One potential explanation for not finding a full alignment between self and other- perceptions of warmth, agreeableness, honesty-humility, and others, can be due to social desirability when providing self-reports. On the other hand, inferring person perceptions over online videoconferencing tools can be an even harder task, compared to in-person conversations. To provide a deeper insight into the reasoning behind observing these results, future research should investigate the impact of context on the accuracy and alignment of person perceptions in social interactions. Similarly, there was no association between how individuals were perceived with their cooperative behavior, while partner selection marginally predicted other's cooperative behavior. Though surprising, this aligns

with prior research showing that people struggle to detect the trustworthiness of strangers [16]. Additionally, the marginal predictiveness of being selected and cooperative behavior may stem from participants not knowing whether their partner had selected them, potentially reducing the reciprocity effect often seen in mutual liking [68].

Partner selection is theorized to have a key role in allowing the assortment of cooperators, thereby promoting cooperation in a social network [6,7]. That said, people may also prefer a cooperative partner whom they could exploit. We found support for the hypothesis that individuals exhibited higher levels of cooperative behavior toward the partners they selected to interact with compared to the partners they did not select. This underscores the significance of partner selection in shaping subsequent cooperative interactions.

## Broader implications

The broader implications of this study are significant for understanding social dynamics in various contexts, including team formation, workplace interactions, and online collaborations, but also for building socially aware intelligent systems. Current findings highlight the importance of warmth and competence in partner selection, especially in task-specific settings. Organizations and teams can benefit from understanding that individuals may prioritize different traits depending on the task at hand. For instance, fostering an environment where warmth and trust are valued can enhance teamwork and cooperation in collaborative tasks, while competence may be more critical in performance-driven scenarios. This application is in line with prior literature showing that in organizations, people tend to select others based on competence, due to their correlation with task performance [69].

Similarly, the usage of automatic annotation of nonverbal behaviors and the usage of machine learning to model social phenomena, such as person perceptions, have two implications. Firstly, automatic annotation was used as a less time-consuming alternative to manual annotation. Moreover, the combination of automatic annotation and machine learning modeling enables us to model complex multivariate temporal relationships and predict perceptions. This helps us to model more complex relationships, but it also highlights the potential of training models that can analyze non-verbal behavior and make predictions of important social phenomena during social interactions. This is important for current efforts in human-AI interaction. For instance, if we can create models that can automatically detect relevant human non-verbal behaviors and use them to reason about how individuals will be perceived in social interactions, we are a step closer to building AI systems that can use these reasons to provide feedback to humans and facilitate better social interactions [65,66,70].

Besides using social science to inform technical fields, innovation in technical fields can help advance social science, specifically social psychology research. Here, we specifically used automatic annotations to show how using machine learning for automatic annotation and data analysis can help capture non-verbal behavioral cues as they emerge in online conversations. Adopting such advancements enables improvements to previous methodologies to investigate social phenomena in their naturalistic settings [19]. Among others [63,64], this study is an example of how such advances can be implemented to contribute towards (a) creating novel approaches for investigating social interactions, and (b) producing research findings that can be used by different communities, such as social psychology and computer science. Nevertheless, machine learning methods are not without their limitations. Thus, more immediate communication between technical and social sciences is needed to further advance this interdisciplinary undertaking.

## Strengths, limitations, and future directions

A strength of this study is the theory-driven approach in designing an experimental task that afforded partner warmth and competence to impact outcomes within an online social interaction setting. The use of online conversations and first-person perspectives enhanced the ecological validity of the findings, which are important for developing applicable systems and interventions. To protect participant privacy, we developed a novel platform for online video conferencing, avoiding third-party data storage.

A limitation of this study is the lack of control over participants' recording environments, such as webcams, microphones, and backgrounds—an inherent challenge in online data collection [71]. Efforts were made to mitigate other online study drawbacks. For instance, participant progress was tracked in real time to minimize distractions. While we ensured participants were fully engaged during online interactions, we cannot guarantee their undivided attention while completing questionnaires, which remains a potential limitation [71]. Additionally, participants chose partners for a single interaction, unlike the repeated interactions typical in real-life scenarios. To enhance contrast, we used two tasks emphasizing warmth and competence. While these tasks served as proxies for real-life scenarios where partner traits contribute to mutual benefit, future research should prioritize testing task affordances in daily life situations. For instance, examining the impact of other's perceived competence and warmth in hiring decisions for roles that afford the expression of these attributes to impact performance and outcomes. Nevertheless, our study produced results consistent with prior research using long-term settings (e.g., hiring scenarios [63]), supporting the validity of our experiment. Another possible limitation is that we relied on pre-trained models (OpenSmile and OpenFace) for feature extraction. Although we used two models due to their wide applicability, future studies could test other models to assess the sensitivity of results to different nonverbal behavior extraction methods.

Additionally, our study did not account for the sex of the partner and the chooser in making partner evaluations and selection, and we did not consider the dynamics of same-sex versus opposite-sex interactions. Future research could explore these aspects, as opposite-sex interactions and gender differences in partner preferences may influence cooperative behavior [72]. Furthermore, we only used one machine learning configuration in our pipeline; future work should compare different configurations to assess their impact on predicting warmth and competence from non-verbal behaviors. Lastly, we focused on a restricted set of non-verbal behaviors, and future work can observe a broader set of non-verbal behaviors, such as posture or eye gaze, while also considering verbal behavior, to assess whether these can enhance prediction accuracy [63].

## Concluding remarks

Past research has shown a strong preference for warm partners across various tasks and relationships. A unique aspect of our study was the experimental manipulation of task affordances for either warmth or competence. Participants met, evaluated, and selected partners for these tasks, showing a general preference for partners perceived as warm and competent. However, they adjusted their selection based on the task, strongly favoring competent partners for the competence task and moral partners (i.e., trustworthiness, a sub-facet of warmth) for the trust task. This suggests that partner selection effectively considers task affordances, aiding in selecting suitable partners across diverse situations.

To date, the study of human behavior within real-life social interactions remains a challenge for social psychologists. This study demonstrates how borrowing tools from other fields, such as the field of social signal processing, can address such challenges in studying social

phenomena in their naturalistic, multimodal, and interactive settings. Specifically, machine learning analyses revealed that acoustic and facial cues predicted perceptions of warmth, but not competence. Competence could potentially be better predicted using other conversational cues, such as communication content. Interestingly, participants' perceptions of others' warmth and competence only partially align with the self-reported traits of the targets. Such discrepancies underscore the need for designing real-life interventions that can promote accurate and effective partner perception and selection. Nonetheless, partner selection was associated with increased cooperation, as participants cooperated more with chosen partners. These data support the idea that partner selection can be an effective mechanism to promote cooperative interactions.

## Supporting Information

**S1 File. Overall supporting information.**
(DOCX)

## Acknowledgments

Special thanks are given to the members of the Socially Perceptive Computing Lab and Amsterdam Cooperation Lab for their comments and insights during presentations. Furthermore, we thank Masha Tsfasman for providing further comments on the manuscript. We thank the reviewers for their insightful comments and suggestions that significantly improved the quality of the current manuscript.

## Author contributions

**Conceptualization:** Tiffany Matej Hrkalovic, Bernd Dudzik, Hayley Hung, Daniel Balliet.

**Data curation:** Tiffany Matej Hrkalovic, Bernd Dudzik.

**Formal analysis:** Tiffany Matej Hrkalovic.

**Funding acquisition:** Daniel Balliet.

**Investigation:** Tiffany Matej Hrkalovic.

**Methodology:** Tiffany Matej Hrkalovic.

**Project administration:** Tiffany Matej Hrkalovic.

**Software:** Tiffany Matej Hrkalovic, Bernd Dudzik.

**Supervision:** Bernd Dudzik, Hayley Hung, Daniel Balliet.

**Visualization:** Tiffany Matej Hrkalovic.

**Writing – original draft:** Tiffany Matej Hrkalovic.

**Writing – review & editing:** Tiffany Matej Hrkalovic, Bernd Dudzik, Hayley Hung, Daniel Balliet.

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
