## [Decision Letter · Decision Letter 0]

19 Nov 2024

PONE-D-24-37688Partner Perceptions During Brief Online Interactions Shape Partner Selection and CooperationPLOS ONE

Dear Dr. Matej Hrkalovic,

Thank you for submitting your manuscript to PLOS ONE. After careful consideration, we feel that it has merit but does not fully meet PLOS ONE’s publication criteria as it currently stands. Therefore, we invite you to submit a revised version of the manuscript that addresses the points raised during the review process.

I apologise for the delayed review process for you manuscript, it was incredibly hard to secure reviewers. First, I’d like to commend the authors on the manuscript. I read the manuscript in detail, and found it to be well-written, answering an important question that is highly relevant to several disciplines in a careful and considered manner. The experimental paradigm is relatively novel, and the treatment and analysis of these quite complex data is appropriate and generates new insights into peer perceptions and partner choice. My enthusiasm is echoed by both reviewers, who recommend minor revisions and several clarifications. Reviewer 1 provides detailed comments and suggestions that I think are important to address during revision. Reviewer 2 notes how impressive the manuscript is and, while they correctly note that there is a lot in the manuscript (and suggest potentially splitting it into two separate manuscripts), I believe that some careful addition to the discussion of the results could ensure that the focus is coherent. Please do, however, respond to all of Reviewer 2's points in your revision. Additionally, I provide some feedback below from my own reading that I think will strengthen the argumentation and clarity presented throughout the manuscript. 

We look forward to receiving your revised manuscript.

Kind regards,

Daniel Redhead

Academic Editor

PLOS ONE

Journal requirements:    When submitting your revision, we need you to address these additional requirements. 1. Please ensure that your manuscript meets PLOS ONE's style requirements, including those for file naming. The PLOS ONE style templates can be found at https://journals.plos.org/plosone/s/file?id=wjVg/PLOSOne_formatting_sample_main_body.pdf and https://journals.plos.org/plosone/s/file?id=ba62/PLOSOne_formatting_sample_title_authors_affiliations.pdf 2. Please include a caption for figure 3. 3. We note that the grant information you provided in the ‘Funding Information’ and ‘Financial Disclosure’ sections do not match.  When you resubmit, please ensure that you provide the correct grant numbers for the awards you received for your study in the ‘Funding Information’ section. 4. Thank you for stating the following financial disclosure:  [D.P (# 864519) awarded from the ERC Consolidator grant (https://erc.europa.eu/apply-grant/consolidator-grant) T.M.H. (funded via Hybrid intelligence Center funded by the Dutch Ministry of Education) grant number 024.004.022 (https://www.nwo.nl/en)].  Please state what role the funders took in the study.  If the funders had no role, please state: ""The funders had no role in study design, data collection and analysis, decision to publish, or preparation of the manuscript."" If this statement is not correct you must amend it as needed. Please include this amended Role of Funder statement in your cover letter; we will change the online submission form on your behalf. 5. We note that you have indicated that there are restrictions to data sharing for this study. PLOS only allows data to be available upon request if there are legal or ethical restrictions on sharing data publicly. For more information on unacceptable data access restrictions, please see http://journals.plos.org/plosone/s/data-availability#loc-unacceptable-data-access-restrictions.  Before we proceed with your manuscript, please address the following prompts: a) If there are ethical or legal restrictions on sharing a de-identified data set, please explain them in detail (e.g., data contain potentially identifying or sensitive patient information, data are owned by a third-party organization, etc.) and who has imposed them (e.g., a Research Ethics Committee or Institutional Review Board, etc.). Please also provide contact information for a data access committee, ethics committee, or other institutional body to which data requests may be sent. b) If there are no restrictions, please upload the minimal anonymized data set necessary to replicate your study findings to a stable, public repository and provide us with the relevant URLs, DOIs, or accession numbers. For a list of recommended repositories, please seehttps://journals.plos.org/plosone/s/recommended-repositories. You also have the option of uploading the data as Supporting Information files, but we would recommend depositing data directly to a data repository if possible. We will update your Data Availability statement on your behalf to reflect the information you provide. 6. We are unable to open your Supporting Information file [Fig1.eps, Fig2.eps, Fig3.eps and Fig4.eps]. Please kindly revise as necessary and re-upload. 7. Please include captions for your Supporting Information files at the end of your manuscript, and update any in-text citations to match accordingly. Please see our Supporting Information guidelines for more information: http://journals.plos.org/plosone/s/supporting-information. 

Additional Editor Comments:

P.3, line 59 onwards: While warmth-competence is a leading theory in social psychology, there are many other (evolutionary) theories on partner choice. Many of these theories of (reputation-based) partner-choice are based on ideas of an individual’s ability and willingness to confer benefits to others but don’t use the terminology “warmth” and “competence” (e.g., Baumard et. al. 2013; Fu et. al., 2008; Roberts 2015). I think that it is important to disentangle ideas of an individual’s ability and willingness to confer benefits from the warmth-competence framework. This could simply be done by having one sentence that outlines that theory points to partner choice being guided by perceptions of an individual’s ability and willingness to cooperate with others (acknowledging the diversity of relevant frameworks), and then follow this with the remainder of the section about warmth-competence being one of such frameworks.

The focus on non-verbal behaviours is necessary, but misses an existing literature on verbal cues (e.g., vocal synchrony: Bowling et. al. 2013; vocal frequencies: Knowles & Little, 2016). This should be acknowledged. Alongside this, there are several examples of how (social) behaviours related to cooperation inform perceptions of an individual’s ability and willingness to confer benefits (e.g., von Rueden et. al., 2019; Redhead et al., 2021)—and partner choice more generally to others in real-world settings (e.g., Redhead, Dalla Ragione & Ross, 2023, Redhead et al., 2024). Many of such studies are not framed around ideas of warmth-competence, but are certainly relevant to the discussion of theory and previous empirical work throughout the introduction.

Please also expand the discussion of limitations related to studies using only online convenience samples, as per Crump et. al. (2013) and Chandler & Shapiro (2016).

Methodology

- Please report the range in age.

- P. 12, lines 253-254: How was the threshold of 2 minutes of conversation decided upon? Was there any systematic patterning of conversations (e.g., were shorter conversations structured by any demographic factors or by combination of the self-report personality traits, and does removing these such conversations significantly change results)? Please elaborate and provide some justification.

Minor notes

- P. 5, Line 92: I think it should be “partners”.

- P. 7, Line 148: The citation for FeldmanHall & Shenhav (2019) seems to be out of place.

- P. 10, Line 218: This should be past tense (“had a conversation”).

- P. 12, Line 264: I think this should be “after providing written consent”.

- P. 28, Line 589: Typo, please remove “in partner selection”.

Suggested References

Baumard, N., André, J. B., & Sperber, D. (2013). A mutualistic approach to morality: The evolution of fairness by partner choice. Behavioral and Brain Sciences, 36(1), 59-78.

Bowling, D. L., Herbst, C. T., & Fitch, W. T. (2013). Social origins of rhythm? Synchrony and temporal regularity in human vocalization. PLoS One, 8(11), e80402.

Chandler, J., & Shapiro, D. (2016). Conducting clinical research using crowdsourced convenience samples. *Annual review of clinical psychology*, *12*(1), 53-81.

Crump, M. J., McDonnell, J. V., & Gureckis, T. M. (2013). Evaluating Amazon's Mechanical Turk as a tool for experimental behavioral research. *PloS one*, *8*(3), e57410.

Fu, F., Hauert, C., Nowak, M. A., & Wang, L. (2008). Reputation-based partner choice promotes cooperation in social networks. Physical Review E—Statistical, Nonlinear, and Soft Matter Physics, 78(2), 026117

Knowles, K. K., & Little, A. C. (2016). Vocal fundamental and formant frequencies affect perceptions of speaker cooperativeness. Quarterly Journal of Experimental Psychology, 69(9), 1657-1675.

Redhead, D., Dhaliwal, N., & Cheng, J. T. (2021). Taking charge and stepping in: Individuals who punish are rewarded with prestige and dominance. Social and Personality Psychology Compass, 15(2), e12581.

Redhead, D., Dalla Ragione, A., & Ross, C. T. (2023). Friendship and partner choice in rural Colombia. Evolution and Human Behavior, 44(5), 430-441.

Redhead, D., Gervais, M., Kajokaite, K., Koster, J., Hurtado Manyoma, A., Hurtado Manyoma, D., ... & Ross, C. T. (2024). Evidence of direct and indirect reciprocity in network-structured economic games. Communications Psychology, 2(1), 44.

Roberts, G. (2015). Partner choice drives the evolution of cooperation via indirect reciprocity. PloS one, 10(6), e0129442.

von Rueden, C. R., Redhead, D., O'Gorman, R., Kaplan, H., & Gurven, M. (2019). The dynamics of men's cooperation and social status in a small-scale society. Proceedings of the Royal Society B, 286(1908), 20191367.

Reviewers' comments:

Reviewer's Responses to Questions

**Comments to the Author**

1. Is the manuscript technically sound, and do the data support the conclusions?

Reviewer #1: Yes

Reviewer #2: Yes

2. Has the statistical analysis been performed appropriately and rigorously? 

Reviewer #1: Yes

Reviewer #2: Yes

3. Have the authors made all data underlying the findings in their manuscript fully available?

Reviewer #1: No

Reviewer #2: No

4. Is the manuscript presented in an intelligible fashion and written in standard English?

Reviewer #1: Yes

Reviewer #2: Yes

5. Review Comments to the Author

Reviewer #1: I very much enjoyed this paper. It is on an interesting topic, will make a clear contribution to the literature, and showcases an interesting methodological and conceptual collaboration between computer science and social psychology. The data also appear to have been collected and analyzed in a very thoughtful, rigorous way.

One risk of such an interdisciplinary collaboration is that you can fall between two stools, so to speak. I come from the (evolutionary) social sciences, and I thought the paper currently has some blind spots with respect to the questions and literature that many readers from my field would find relevant to this research. What follows then are some suggestions for making the paper clearer and more comprehensive, especially focused on enhancing the framing of the paper from an evolutionary/social psychology perspective.

1. The potential effects of participant sex and partner sex are not addressed at all. This seems like an obvious oversight for a paper on partner choice and cooperation. In opposite-sex interactions, you might expect partner attractiveness to affect rates of partner selection and cooperativeness in the trust game (e.g. Maestripieri et al., 2017, Behavioral and Brain Sciences). You might also expect participants (especially men) to be more cooperative in the trust game with opposite-sex vs. same sex participants, and there’s evidence that men and women behave differently in same-sex economic games (e.g. men might be engaged in more partner choice/attraction, while women might be more intrasexually competitive; e.g. Eisenbruch et al., 2016, Evolution & Human Behavior; Lucas & Koff, 2013, Evolution and Human Behavior). To the extent possible, it would be interesting to explore the effects of participant sex, whether the trial was same- or opposite-sex, and their interaction. At a minimum, this topic should be discussed as a limitation and/or direction for future research.

2. The fact that people adjust their preferences for warmth and competence to the task type is interesting. It should be connected to the existing literature on relationship-specific social preferences (e.g. Sprecher & Regan 2002, JSPR; Cottrell et al., 2007, JPSP).

3. At the same time, it’s very interesting the degree to which people did NOT adjust their preferences to the task type. While people’s preference for competence only existed in the joint competence task (very clever task, by the way), the preference for warmth was present in both conditions. In other words, even when the task had no affordance for warmth to affect outcomes at all, people still had a pretty strong preference for warm partners. This is consistent with the existing literature on the primacy of warmth over competence in social perception and partner choice (e.g. Fiske et al., 2007, TiCS; Eisenbruch & Krasnow, 2022; Raihani & Barclay 2016; Robertson, T. E., Krasnow, M., & Lim, J. (2017, May 31–June 3). People prefer those who value them over those who benefit them [Paper presentation]. 29th Annual Human Behavior and Evolution Society Conference, Boise, ID, United States.). The fact that people did this when it was unambiguously irrelevant to the task at hand suggests that people’s partner preferences are partially rationally calibrated to their current situation, partially hardwired (perhaps by evolution).

a. The effect of warmth on partner selection in the joint competence task is misstated on p. 32, lines 682-684. What’s written here contradicts the results reported on p. 25 and Figure 3.

4. It’s interesting that participants’ competence perceptions tracked the partners’ performance in the cognitive task in phase 1 of the study. Did the perceptions of competence track actual performance in the joint competence task? There is a small but growing literature on the ability to accurately detect certain forms of competence from (men’s) faces (e.g. IQ: Kleisner et al., 2014, PLOS ONE; hunting ability: Eisenbruch et al., 2024, EHB; fighting ability: Zilioli et al, 2014, Aggressive Behavior), which is consistent with the accuracy of competence perceptions reported here.

5. The point about the need for social preference studies based on actual interactions (instead of single cues like just faces or voice recordings) is a good one, but there are a small number of existing studies that do that. Eisenbruch et al., 2019 (Adaptive Human Behavior & Physiology) had participants interact face-to-face and then play economic games, so it seems relevant to the current study. There is also a literature based on the “fast friends” procedure; Sprecher 2021 (JSPR) might be relevant.

6. In addition, there’s research on the inferences made from people’s voices and the effects of vocal parameters on social decisions. This would provide good theoretical grounding for analyzing the vocal parameters recorded from the conversation. For example, Aung et al., 2024 (Psychological Science) is a good, recent cross-cultural study of social perceptions based on voice pitch, and there’s a bunch of research on voice pitch affecting election outcomes (e.g. Tigue et al., 2012, EHB). Among hunter-gatherers, men with deeper voices are actually better hunters (Smith et al., 2017, Evolutionary Psychology), so there’s a plausible evolutionary account of why this would be relevant to partner choice.

7. The passage on p.5 lines 104-111 could use a bit more explanation. It’s not clear immediately what you mean by manual annotation and automatic annotation.

8. The stuff about the actor-observer effect (e.g. p. 6 lines 126-128) seems like a bit of a stretch and not really necessary. People judging their conversation partner are still an observer with respect to their partner. This part of the framing, and testing the relationship between participants’ perceptions of their partners and the partners’ self-perception, was the least interesting to me. I’d suggest dropping it or maybe moving those results to SI – there’s plenty else of interest here that this experimental design can speak to more directly.

9. On p. 10 lines 213-220, make it clearer that everything is happening online.

10. p. 16 lines 340-343: That’s a sloppy sentence. Please make it clearer.

11. p. 20 line 416-p. 22 line 460 should be moved to the Methods section.

12. p. 24 line 495: what do you mean by batch size?

13. p. 26 line 546: are those the reported statistics for an interaction effect? I think I figured it out from context, but it’d be better to be more explicit about what is being tested and reported in each result.

14. p. 27 line 580: By cooperative behavior, you mean amount sent in the trust task, right? I think little reminders like this would help the reader understand everything – there’s a lot going on here.

15. p. 28 line 586: significant predictor of what?

16. p. 28 line 589: “the role of partner selection in partner selection” – please proofread and fix.

17. p. 28 line 609-p. 29 line 610: I think you should make clear that the automatic annotations of facial and vocal cues could weakly predict perceptions of warmth. The vast majority of the variance in warmth perceptions was unaccounted for by the machine learning algorithms. It’s cool that automatic annotation predicted anything at all, but I think that some perspective regarding the effect size is necessary here.

18. p. 30 lines 636-650 discusses the cues that were and weren’t present in the conversation, but omits what the participants actually said. I think this should be mentioned as the likely explanation for most of the variance in warmth and competence perceptions, and the contents of the conversation would make a good target for future research. (Maybe expand on the mention of “verbal information” in line 664.)

19. p. 33 lines 709-710. It’s a bit strange that warmer partners were not more cooperative but selected partners were more cooperative, given that warmer partners were more likely to be selected. It seems like warmth promotes selection, and selection does predict cooperativeness, but the warmth � cooperativeness relationship is not independently significant. I think perhaps this could be interpreted with a bit more nuance than just saying that partners perceived as warmer were not more cooperative.

I hope that some of these comments will be helpful for you. I understand that space constraints and the need for the manuscript to have some discernable focus might preclude addressing all of these points, but generally I think the manuscript could use a stronger grounding in the relevant social and evolutionary psychology literatures. Pending that revision, I think it’s going to make a nice contribution to the literature.

Reviewer #2: This manuscript presents an exploration of person perception and partner choice. I’m embarrassed to say I don’t have much constructive to say here. The aims are clear, important, and target well-defined gaps in the literature. The methodology is inventive and impressive: I am impressed with the thorough data collection, the use of two behavioral tasks that separately isolate competence and warmth, as well as the effort to automatically annotate live interaction data—all with Prolific participants! I can see this study laying a useful foundation for future work on partner choice. The results are interesting and compelling: that perceptions of competence and warmth from real interactions independently predict partner choice for the distinct tasks is quite interesting.

My only slight criticism is that the aims of the manuscript feel a bit diffuse. It feels to me that there are two manuscripts here: one on partner choice and one on person perception. As a person admittedly more interested in the partner choice elements here, although technically very impressive, I don’t see as much value added by the automated annotation results. If we know that people are forming perceptions of warmth and competence from their interactions with their partners and we know that these perceptions are guiding their choice behaviors, what do I learn from knowing that a machine learning model can also predict some of the variance in some of the perceptions from the interaction data? This is, I concede, somewhat of an aesthetic choice: I’m sure a researcher interested more in person perception would find the annotation results and the perception accuracy results more interesting than the partner choice results. It seems to me that these could be two different manuscripts, with each set of results given the space and attention they deserve. Or one set of results could be backgrounded to supplementary material. However it is resolved, these distinct foci at present don’t really cohere into a single manuscript for me at the moment.

On a very minor note: I think there may be an issue with the OSF page settings as it appears empty when I try to access it.

6. PLOS authors have the option to publish the peer review history of their article (what does this mean?). If published, this will include your full peer review and any attached files.

Reviewer #1: No

Reviewer #2: No

---

## [Author Response · Author response to Decision Letter 1]

20 Dec 2024

All of the responses to specific reviewers and editor comments can be found in the file Response to Reviewers. As we have colors and italized text that is not presented here, we think that in the document we were able to more appropriately address the comments.

---

## [Editor Report · Decision Letter 1]

12 Jan 2025

Partner Perceptions During Brief Online Interactions Shape Partner Selection and Cooperation

PONE-D-24-37688R1

Dear Dr. Matej Hrkalovic,

We’re pleased to inform you that your manuscript has been judged scientifically suitable for publication and will be formally accepted for publication once it meets all outstanding technical requirements.

Kind regards,

Daniel Redhead

Academic Editor

PLOS ONE

Additional Editor Comments (optional):

The authors have provided detailed responses, appropriate revisions and convincing justifications for all points raised by the reviewers and myself. I believe that the paper is of excellent quality, and is now in a state that is acceptable for publication.
---

## [Editor Report · Acceptance letter]

PONE-D-24-37688R1

PLOS ONE

Dear Dr. Matej Hrkalovic,

I'm pleased to inform you that your manuscript has been deemed suitable for publication in PLOS ONE. Congratulations! Your manuscript is now being handed over to our production team.

Kind regards,

on behalf of

Dr. Daniel Redhead

Academic Editor

PLOS ONE